# Thermoregulation during Field Exercise in Horses Using Skin Temperature Monitoring

**DOI:** 10.3390/ani14010136

**Published:** 2023-12-30

**Authors:** Elisabeth-Lidwien J. M. M. Verdegaal, Gordon S. Howarth, Todd J. McWhorter, Catherine J. G. Delesalle

**Affiliations:** 1Thermoregulation Research Group, Equine Health and Performance Centre, University of Adelaide, Roseworthy, SA 5116, Australia; gordon.howarth@adelaide.edu.au (G.S.H.); todd.mcwhorter@adelaide.edu.au (T.J.M.); catherine.delesalle@ugent.be (C.J.G.D.); 2School of Animal and Veterinary Sciences, Roseworthy Campus, University of Adelaide, Roseworthy, SA 5371, Australia; 3Research Group of Comparative Physiology, Department of Translational Physiology, Infectiology and Public Health, Faculty of Veterinary Medicine, Ghent University, 9820 Merelbeke, Belgium

**Keywords:** thermoregulation, body temperature monitoring, skin temperature (T*_sk_*), infrared thermography (IRT), sports medicine, exercise, horse, equine, core temperature (T*_c_*), continuous monitoring

## Abstract

**Simple Summary:**

Heat stress is an important performance and welfare issue for exercising horses. The process of thermoregulation is crucial for equine athletes. Excessive metabolic heat generation during exercise, combined with inefficient heat dissipation, can lead to hyperthermia if not detected in time and not effectively managed. Accurately monitoring heat generation during exercise allows for early preventative intervention as body temperature rises. Skin temperature monitoring is commonly used as a non-invasive method to assess body temperature responses pre- and post-exercise. To date, few studies have used infrared thermographic techniques to monitor body temperature continuously during exercise under laboratory conditions and in the field. In reviewing these results, the accuracy of measuring skin temperature as a reliable indication of overall body temperature is discussed. This commentary summarizes the results of studies measuring surface skin temperature in horses, particularly using infrared thermography for exercise-focused monitoring in the field.

**Abstract:**

Hyperthermia and exertional heat illness (EHI) are performance and welfare issues for all exercising horses. Monitoring the thermoregulatory response allows for early recognition of metabolic heat accumulation during exercise and the possibility of taking prompt and effective preventative measures to avoid a further increase in core body temperature (T*_c_*) leading to hyperthermia. Skin temperature (T*_sk_*) monitoring is most used as a non-invasive tool to assess the thermoregulatory response pre- and post-exercise, particularly employing infrared thermographic equipment. However, only a few studies have used thermography to monitor skin temperature continuously during exercise. This commentary provides an overview of studies investigating surface skin temperature mainly by infrared thermography (IRT) during exercise. The scientific evidence, including methodologies, applications, and challenges associated with (continuous) skin temperature monitoring in horses during field exercise, is discussed. The commentary highlights that, while monitoring T*_sk_* is straightforward, continuous T*_sk_* alone does not always reliably estimate T*_c_* evolvement during field exercise. In addition, inter-individual differences in thermoregulation need to be recognized and accounted for to optimize individual wellbeing. With the ongoing development and application of advanced wearable monitoring technology, there may be future advances in equipment and modeling for timely intervention with horses at hyperthermic risk to improve their welfare. However, at this point, infrared thermographic assessment of T*_sk_* should always be used in conjunction with other clinical assessments and veterinary examinations for a reliable monitoring of the welfare of the horse.

## 1. Introduction

Field exercise is an integral aspect of equine sports and competition where horses face a myriad of physiological and environmental challenges. Efficient thermoregulation is paramount to ensure optimal performance and prevent heat-related stress in equine athletes. The incidence of heat stress and exertional heat illness (EHI) is projected to rise further due to the ongoing effects of global climate change. Climate change is a generally accepted phenomenon, and the scientific community is warning the world about its long-term consequences both for human and animal health [1,2,3]. Particularly, more acute changes in environmental conditions such as heat waves, entail that athletes may need to perform under circumstances to which they are not acclimatized. These sudden increased ambient temperatures (T*_a_*) are the dominant external risk factor for heat stress for human and equine athletes [1,2,3].

EHI is associated with various kinds of exercise and impacts the overall well-being of all mammals, including humans, and may lead to exertional heat stroke (EHS) and ultimately death. In exercising human athletes, EHS is among the top three causes of sudden death, and in summer, it is the number one cause of athlete death in the USA [4]. For example, EHI is common in American football, with an average incidence rate ranging from 0.01 to 0.4%. Heat monitoring studies being developed to investigate the EHI problem in human athletes have used many monitoring approaches such as ingestible thermo-monitoring pills or a rectal thermistor, skin temperature (T*_sk_*) sensors, and measures of subjective perception of heat load [5].

Both EHI and EHS are also problematic conditions in equine athletes and can occur during a wide variety of different types of exercise [6]. However, heat exhaustion triggered by thermoregulatory-induced physiological feedback failure is most commonly reported when horses need to perform submaximal exercise during extended time intervals such as endurance rides. The prevalence of metabolic disorders in endurance horses ranges from 2% to 15% [7,8,9,10,11,12,13,14]. In comparison, for maximal intensity exercise, EHI screening studies have recently reported a prevalence between 0.1% and 1.1% in racehorses in Australia, the UK, and Japan [15,16,17,18,19]. A study in eastern Australia focused on selected EHI cases post-exercise at the racetrack and suggested an EHI incidence of up to 9.5% during hot summer months [17]. The latter study used four severity levels of EHI (1 = least severe) reported by Brownlow et al. [20] and concluded that 96% of horses could be categorized as level 1. This suggests that many EHI cases of low-level severity with vague clinical signs may have been overlooked in the past. At present, the association between EHI and exercise injuries leading to fatalities is not well understood, and hence the roles that heat exhaustion and EHI play are unknown.

All equestrian sports are facing growing scrutiny from both the general public and animal welfare organizations. When unfortunate incidents result in fatalities, spectators are swift to criticize sporting and racing events and increasingly call for safeguarding the well-being of sports horses in order to maintain the “social license” to operate in equestrian sports [21]. Consequently, from an animal welfare perspective, it is paramount and timely to prevent occurrence of heat stress on all occasions. The aforementioned welfare issues are a concerning reality that is motivating research groups worldwide to develop practical methods for reliable monitoring during field exercise to ensure the thermoregulatory welfare of horses [21,22,23,24,25,26]. To achieve this, there is a need for minimally invasive body temperature monitoring systems, also termed ‘heat monitoring’. The ultimate goal of monitoring is to accurately detect a potentially dangerous rise in core body heat in real-time such that effective preventative actions can then be taken.

In an ideal world, all temperature monitoring equipment including wearable technological tools should be non-invasive and enable horse owners and veterinarians to have a continuous real-time view of the evolution of body temperature during exercise in the field [22,23,24,25]. An approach that is gaining popularity for monitoring the thermoregulatory response is the follow-up of the T*_sk_* in horses performing in the field exercise using wearable T*_sk_* equipment [24]. While this equipment provides output of physiological data, the question remains whether these data have physiological relevance for the evolution of the core body temperature, which is the core point of interest to safeguard welfare in a reliable way. This article reviews and comments on the benefits and challenges of monitoring thermoregulation in exercising horses using non-invasive surface T*_sk_*, particularly through infrared thermography.

## 2. Thermoregulation: Core vs. Shell Temperature

Thermoregulation is the internal physiological process which balances metabolic heat load and the exchange of this heat with the environment. Overall body temperature in mammals is divided into an inner core body temperature (T*_c_*) and an outer shell temperature. T*_c_* in horses is regulated within a narrow range (37.4–38.0 °C), while shell temperature varies more widely in response to ongoing central thermoregulatory processes [24,26]. The inner core temperature refers to deep-body temperatures, while shell temperature includes intramuscular, subcutaneous, and surface skin temperatures (T*_sk_*). Heat is moved between core and shell areas to equilibrate heat production and heat loss and thus regulate the overall body temperature. Consequently, temperature is not uniform at different locations in and on the body. Thermoregulatory processes are controlled via neurophysiologic control centers in the hypothalamus. Both peripheral and central thermoreceptors are nerve endings that respond to changes in T*_c_* and T*_a_*, all in communication with the hypothalamus (Figure 1). The peripheral thermoreceptors in the skin detect the peripheral shell temperature and respond to a range of external temperatures from cold to hot (5–60 °C), with cold thermoreceptors being the most numerous.

## 3. Mechanism of Metabolic Heat Loss

During exercise, heat originating from exercising muscle groups will be dissipated to the surrounding environment through different tissues and by using transportation via the bloodstream towards the skin surface. *H* will then be released into the surrounding environment through four mechanisms: radiation, conduction, convection, and evaporation (Figure 1). Therefore, the surface area (cm^2^) of the body is a critical factor in determining rates of heat exchange. Because heat exchange with the environment is proportional to the relative size of the body surface area (BSA), the rate of heat exchange per unit of body mass (BM) is largest in the smallest animal (if other variables are equal). For example, the BSA-to-volume ratio is 50% less in a 500 kg horse when compared to humans [26,27]. This lower BSA-to-BM ratio of horses results in greater demands being imposed on their thermoregulatory system during exercise [28,29,30]. Only if T*_sk_* is higher than the surroundings can the body lose heat by radiation, conduction, and evaporation.

Among these heat-loss pathways, the most critical and pivotal for horses is the activation of sweat glands to evaporate sweat from the skin surface [27,29,31,32,33,34]. The efficacy of heat dissipation via this evaporative process relies on the thermal contrast between locally perfusion-based T*_sk_* and its immediate surroundings, with consideration of variables such as vapor pressure, airflow, and solar radiation, especially during outdoor exercise [17,28,35] (Figure 1). Consequently, when considering using T*_sk_* as a method for monitoring temperature, it is essential to anticipate that all pathways for dissipating *H* to the environment can influence the T*_sk_* data output.

Additionally, horses are hindgut fermenters, which entails that important fermentation processes take place within the gastro-intestinal (GI) system [26]. These processes are recognized for their propensity to produce an enormous amount of heat which in turn represents an additional challenge for the thermoregulatory system. It is quite plausible that the type of dietary load (composition of intestinal content) inside the GI system of a horse performing exercise has its impact on T*_c_* evolution at that time point [26]. How quickly the latter is translated towards the outer shell temperature is currently unknown, but again, requires notice as a factor that may influence the T*_sk_* data output. From a comparative perspective, in humans it is known that food containing high amounts of fats, complex carbohydrates, and proteins stimulate the T*_c_* increase during digestion [26].

## 4. Field-Based Thermoregulation Research—Non-Invasive Monitoring

Until recently, little was known about how thermoregulation functions in real-life circumstances during field exercise and recovery in different equine sports disciplines. An important reason for this was the complete lack of effective, reliable temperature monitoring equipment that can be safely and comfortably applied during field exercise. In the past, most equine thermoregulatory research has been performed either under laboratory conditions, for example on a treadmill [36,37,38], or under field conditions, only focussed on the pre- and post-exercise period. A few studies have aimed at continuous monitoring of the T*_c_* during exercise and recovery in real-time field training and competitions. One of those studies used intra-uterine temperature loggers as a nonsurgical, minimally invasive method [39], an approach which only allows for monitoring in female horses. Recently, a reliable novel method is the oral administration of a small telemetric GI temperature pill to monitor T*_c_* on a continuous basis [40,41,42,43].

The aforementioned field studies reveal that in-the-field exercise monitoring is essential to obtain a holistic view on how the thermoregulatory system actually works. This is not only challenged by the performed exercise itself, but also by the environmental conditions in which environmental temperature and humidity are pivotal factors [43]. Additionally, the terrain itself influences exercise intensity. For example, the presence of slopes and descents in a competition course as well as varying properties of the surface over the course, such as sand or mud, are challenging for optimal exercise capacity [43]. These factors must all be accounted for when monitoring T*_c_* during exercise, and for that purpose, monitoring thermoregulation during field exercise has been shown to be more reliable than lab-based testing [43]. In addition to thermoregulation monitoring, it is important to consider thermal comfort when assessing the welfare of horses (Figure 1). Thermal comfort refers to a state in which horses can still successfully thermoregulate, but may already feel thermally uncomfortable. This is similar to evaluating a perception of heat exertion during heat monitoring in human athletes. Many factors that impact thermal comfort, such as solar radiation, skin insulation, and wind speed, can also affect T*_sk_* values.

## 5. Skin Temperature Measurement

Non-invasive methods such as T*_sk_* measurement are of significant interest and have found various applications. Generally, four types of temperature sensor equipment are employed in thermoregulation research: thermometers, thermistors, thermocouples, and infrared thermography (IRT) devices. Of these, IRT has received the most extensive attention in recent equine studies and is commonly used remotely (non-contact temperature sensor) [22,44,45,46,47,48,49,50,51,52,53,54,55,56]. Digital temperature sensors may include technology that comprises a logger consisting of a temperature sensor in a stainless-steel computer chip with an enclosed battery [57]. Thermistors are temperature-sensitive resistors that change resistance with temperature and have been used in several equine studies [34,58,59,60]. Both digital temperature sensors and thermistors have a small size, fast response time, and relatively low cost. However, they are invasive, are limited to point measurements, and may alter the local skin conditions. Thermocouples are temperature sensors based on the Seebeck effect, where a voltage is generated in response to a temperature difference (thermo-electric sensors) and have been used in some equine studies [57,61,62]. In addition, fiber optic sensors can be used to record temperature by measuring the change in light transmission properties through an optical fiber to measure temperature [63]. In human thermoregulatory research, conductive textiles have received increasing attention recently. These are textiles with embedded conductive fibers that can be used to measure T*_sk_*. However, this technique still encounters many calibration challenges, and the accuracy of data output is influenced by textile placement [63]. In addition, the dermal patch technology is a small adhesive patch with integrated sensors applied to the skin surface [63]. Lastly, IRT measures the infrared radiation emitted by an object to create a temperature map [22] and has multiple applications in equine medicine, including detecting inflammation and musculoskeletal injuries such as lameness, tendonitis, and laminitis [51,64,65]. In several studies, IRT has been used as an indicator of stress, but those studies investigated only a limited number of horses (five horses) [44,46,65,66] and one study investigated the correlation of increased T*_sk_* with blood lactate concentration post exercise in 30 horses [67]. Lastly, infrared contact sensors (IRCSs) are contact-based sensors using infrared technology to measure T*_sk_* [24,60].

## 6. Skin Temperature at Point(s)-in-Time or Continuous Monitoring during Exercise

Changes in T*_sk_* during field exercise are complex. As exercise continues, T*_sk_* varies by body region and may be transient and rapidly changing in relation to local heat exchange from the muscles to the environment, an increased sweat rate, and forced convection (wind), all highly influenced by T*_a_*.

### 6.1. T_sk_ Measurement at a Point-in-Time (Pre-Exercise and Post-Exercise)

The most common method to record T*_c_* in field competition is serial measurement of the rectal temperature (T*_re_*) pre- and, more often, only post-exercise. These are “snapshots” (points-in-time) of temperature evolvement. Importantly, research has shown that T*_re_* lags behind or is always lower than T*_c_* [28,42]; consequently, serial T*_re_* measurement does not allow for early-stage remedial intervention. Importantly, there is also a clear lack of correlation between T*_sk_* and T*_c_* [24]. To date, several methods are employed for monitoring T*_sk_* such as infrared thermography using cameras. While handheld IRT cameras are becoming increasingly popular in equine sports medicine due to their non-invasiveness, most exercise studies involve point-in-time T*_sk_* measurements only post-exercise and without pre-exercise and intra-exercise periods [22,30,45,46,47,48,52,53,58,68,69,70]. A non-exhaustive overview of T*_sk_* studies associated with exercising horses, either monitoring non-continuously or continuously in horses, is presented in Table 1.

A systematic review of 45 T*_sk_* studies of human endurance athletes reported that only a few studies involved continuous temperature monitoring [23]. From a physiological viewpoint, it is more than plausible that not only the actual temperature increases, but also the rate of increase and the time profile which are important parameters to reliably estimate how an individual athlete copes with heat load. Those parameters remain a “black box” when only non-continuous temperature monitoring is applied. Some human studies have applied highly frequent monitoring intervals. Two studies monitored T*_sk_* every 30 s, one study every 180 s, and four studies every 5 min [23]. These human studies highlighted essential inter-individual differences concerning the T*_sk_* response time profile [23]. Similarly, significant inter-individual T*_sk_* differences have been reported in all mammals [33], including a recent equine exercise study [24]. In addition, T*_sk_* in 21 horses monitored at rest also revealed significant inter-individual differences, which entails that defining “thermo-comfort” reference ranges actually needs a customized approach [50].

In horses, several studies have revealed possible correlations between recorded point-in-time T*_sk_* values and various other physiological parameters [24,45]. For example, a study involving eight endurance horses analyzed the association between endurance training intensity and T*_sk_* evolvement using IRT cameras focused at different locations on the body. This revealed that T*_sk_* at the coronary band increased with training intensity, but maximum T*_sk_* did not [45]. Another study monitoring endurance horses revealed that a higher T*_sk_* at the end-of-exercise period was associated with a lower T*_c_* at the end-of-recovery period (60 min) [24]. In addition, there was no significant correlation between T*_sk_* at the end of exercise and the duration of HR recovery to 60 bpm [24]. Several hypotheses could be considered to explain the marked difference between T*_sk_* and T*_c_*, both with respect to temperature values and evolvement over time. For example, it could be hypothesized that the raised T*_sk_* functions as an “early whistle blower” for the launch of an active thermoregulatory response to anticipate the increased T*_c_*, and once the metabolic heat is successfully dissipated, the T*_c_* decreases. This argument could be coupled with the effect of cooling-down measures being applied post-exercise, which may be more prominent when T*_sk_* is greater and ultimately result in higher dissipation of metabolic heat and swifter reduction of T*_c_*. However, the important fact that both temperature values and time evolvement patterns importantly differ between T*_sk_* and T*_c_* shows that the physiological link between both is much more complicated than initially anticipated in studies investigating the many monitoring wearables on the equine market.

Interestingly, an immediate post-race, high point-in-time IRT T*_sk_* measurement > 39 °C in the neck region of racehorses exercising in a warm-hot T*_a_* has been advocated as an early indicator for increased risk for development of post-exercise EHI [48]. Possibly in the case of the manifestation of a narrow temperature gradient between T*_c_* and T*_sk_*, the capacity of the equine body to transfer heat to the skin is greatly reduced and thus most probably hinders heat dissipation through evaporation [24,48]. Therefore, additional research has tested the effectiveness of different cooling-down techniques with the obvious goal of increasing the temperature gradient between the T*_sk_* and the T*_c_* post racing. A recent study evaluated five cooling methods for racehorses—walking with and without air fans, intermittent application of cold water with or without scraping, and continuous pouring of tap water over the body [52]. The study revealed a mean T*_sk_* of 40 °C at the end-of-race exercise in a warm environment (mean T*_a_* 31.8 °C) [52] and a T*_sk_* lower than 34.5 °C at the end of moderate intensity exercise in a moderate environment (mean T*_a_* 24 °C) [59]. Of considerable concern is the occurrence of a markedly increased T*_sk_* post-exercise (>39 °C) which could indicate the horse is potentially already manifesting mild to severe manifestations of EHI.

Significantly, all of the aforementioned studies were performed under a wide range of T*_a_*, from hot to cool conditions, and tested large disparities in exercise intensity ranging from racing to endurance exercise [24,45,48]. The pattern of T*_sk_* and its variability can partly be attributed to acute fluctuations in blood flow associated with differing exercise intensities [32,60]. In racehorses executing high-intensity, short-duration exercise, heat dissipation will typically occur post-exercise, while endurance horses manage heat dissipation throughout their submaximal prolonged exercise and their mandatory monitored rest periods [43,48,71,72].

### 6.2. Continuous T_sk_ Monitoring during Exercise

To date, study results have clearly shown that the relationship between T*_sk_* and T*_c_* is very complex [32,33,72], and those studies that involve continuous temperature monitoring are pivotal to better understand this complexity. Only eight studies have applied continuous T*_sk_* monitoring during exercise, of which four studies involved a controlled laboratory environment using treadmills [36,37,38,44] (Table 1). These laboratory-based investigations cannot address the authentic environmental conditions encountered by sports horses during open-air competitions. By comparison, different important variables can be deduced from the analysis of the IRT data time profiles during exercise in the field such as the average, maximum and minimum T*_sk_*, standard deviation, and delta T*_sk_* (T*_sk_* variation during the exercise) [24].

Currently, only a few studies have tried to correlate both continuous T*_c_* (through either arterial, venous blood, or GI pill temperature) and T*_sk_* in an attempt to analyze if T*_sk_* could function as a proxy for the core body thermoregulatory response [24,36,37,38,62] (Table 1). Importantly, the four laboratory-based treadmill investigations consistently affirmed the absence of a significant correlation between T*_sk_* and T*_c_*. For instance, two submaximal exercise studies used arterial blood temperature as a reference T*_c_* to compare the impact of different environmental conditions varying from cool, dry (room T*_a_* and 45% relative humidity (RH)) to hot, humid (34 °C, 85% RH) on the thoracic surface T*_sk_* using thermocouples and revealed dissimilarities between T*_sk_* and T*_c_* [61,62]. Two high-intensity exercise studies investigated tail surface T*_sk_* evolvement response to cooling-down techniques (6 times cold water shower for 30 s) and acclimation (30 °C and 80% RH over 15 days), respectively, and both demonstrated clear distinctions between T*_sk_* and T*_c_* recordings [37,38]. One other study did not use T*_sk_* monitoring but recorded muscle temperature using intra-muscular microchips under laboratory conditions [73]. In that study, the muscle temperature continuously increased and lagged behind the central venous temperature during the recovery phase. This non-alignment is in accordance with previous studies reporting on the lack of direct correlation between muscle temperature and T*_c_* [29,74,75].

Only four studies have applied continuous T*_sk_* monitoring during field exercise and thus avoided laboratory conditions [24,57,61,62] (Table 1). None of these studies has identified a correlation between T*_sk_* and T*_c_*. In a recent study, we investigated the possible correlation of T*_sk_* with T*_c_* involving 13 endurance horses in a real-life competition [24]. The T*_c_* was continuously measured using an ingested telemetric GI pill, and T*_sk_* was continuously recorded every 15 s using an infrared thermistor sensor integrated into a modified belt during an endurance ride. Our study revealed no significant correlation between the profiles of T*_sk_* and T*_c_* during exercise and recovery phases, which demonstrated that monitoring T*_sk_* does not reliably serve as a proxy for assessing the important core body thermoregulatory response during field exercise. Additionally, our study highlighted significant inter-individual variations in T*_sk_* and T*_c_* profiles, emphasizing the importance of implementing an individualized temperature monitoring model for all sport horses. Our previous field study [43] confirmed a substantial inter-individual variability in the T*_sk_* time profiles despite the same exercise protocol. Similar findings have been reported in human athlete studies [76,77,78,79]. Consequently, researchers have developed integrative models that utilize T*_sk_* to assess heat balance during exercise. This approach has been explored in both human studies and one equine study [80,81,82]. However, in a field study involving 17 human marathon runners, no regression model could effectively predict physiological heat stress load using single-point IRT T*_sk_* measurements [83]. It is important to consider that this lack of reliable prediction of heat load can also apply to exercising horses in the field.

## 7. Overview of Factors Influencing the T*_sk_* Measurement

While efforts are being made to incorporate continuous T*_sk_* monitoring into field-based thermoregulation assessment, researchers need to be fully aware of several factors that add to the complexity of the physiological relationship between T*_sk_* and T*_c_*. Skin temperature evolvement depends on several individual equine variables such as breed and genetic factors, sex, age, body composition, coat color, and physical fitness level. These factors are coupled with environmental parameters (such as T*_a_*, RH, and wind speed), attributes of the exercise (duration and intensity, additionally influenced by terrain conditions), and the possible additional application of cooling-down interventions. Notably, T*_sk_* represents the outer shell temperature and thus is much more easily influenced by additional factors when compared to the T*_c_*. From a physiological viewpoint, the T*_c_* should be guarded with much greater scrutiny and precision when compared to the outer shell temperature. It is critical to remember that every monitoring device relies on a specific type of technology, each having advantages and shortcomings. Therefore, the choices of the type of temperature sensor equipment as well as the specific anatomical location of the T*_sk_* measurement need to be taken into account throughout the studies (Table 1).

### 7.1. Environmental Factors

Evidently, the added value of performing field studies is the capacity to involve the effect of environmental factors on thermoregulation in exercising horses. Due to climate change, some environmental conditions can worsen quickly, pressing highly trained horses to perform under sudden changed climate conditions (such as heat waves) without adaptation and acclimatization.

Evaporation is the endothermic process in humans and horses during which liquid (sweat) transforms into vapor. The capacity to sweat relies on an active vasodilatory and vasoconstrictor mechanism controlled by the autonomic nervous system in horses during exercise [26,32,84]. Evaporation of sweat also depends on the skin-to-ambient-vapor pressure difference as well as on the T*_sk_* to T*_a_* difference. Because of the dependence of the evaporation process on the temperature and the vapor gradient between the skin and the immediate surroundings, this process will have a reduced capacity in hot and humid weather. Environmental factors play a substantial role in shaping the dynamics of T*_sk_* and T*_c_* despite different exercise intensities and their interplay, as these external factors have the capacity to swiftly modify T*_sk_* without directly influencing T*_c_* [33,68,85,86,87,88]. These factors include T*_a_*, solar radiation, soil radiation, RH, shade, and wind speed (vs. enclosed areas without wind). For instance, fluctuations in T*_a_* within the range of 20 °C to 30 °C have been associated with the initiation of skin vasodilation and sweat evaporation [89]. Solar radiation, by heating the skin, may contribute up to 15% of the total heat gain in an exercising horse depending on local air velocity (wind speed). In one study, T*_sk_* and T*_re_* reached their peaks during maximum solar radiation in daytime hours [87,90].

Exercise studies conducted in warmer environments have documented post-exercise T*_sk_* measurements surpassing 39 °C. For example, a recent study found that 28 out of 38 horses exercising in a hot, dry setting (mean T*_a_* 38.8 °C) and 6 out of 37 horses exercising in a warm, humid environment (mean T*_a_* 31.1 °C) exhibited post-exercise T*_sk_* values exceeding 39 °C. The researchers suggested that horses with T*_sk_* recordings exceeding 39 °C are at risk of heat stress and EHI, emphasizing that this T*_sk_* response may identify racehorses in need of cooling down [48]. Verdegaal et al. [24] found that the mean T*_sk_* for endurance horses after exercise in a cool environment (mean T*_a_* 15.3 °C) was 28 °C, and none of the horses had a T*_sk_* exceeding 39 °C. It is important to note that in the latter study, the T*_sk_* sensor was placed on the chest and protected from solar radiation by a belt and girth which prevented it from being influenced by environmental factors like solar radiation.

### 7.2. Individual Equine Factors and Thermoregulation

Horses have several individual factors that may impact their thermoregulation, including breed, body condition score, age, and temperament (including nervousness). Various skin-related factors such as sweating rate, skin thickness, blood vessel density, hair coat characteristics, presence of hair, clipping practices, coat color, and anhidrosis, all influence the efficacy of their sweat evaporation and the T*_sk_* measurement [27,30,33,53,54,56,68,80,91,92]. The influence of breed on T*_sk_* relates to the BSA-to-BM ratio, where a higher ratio leads to increased heat dissipation [89]. Clipping of the hair coat resulted in reduced T*_sk_* and T*_re_* [30,54,68]. Additionally, coat color may hold some relevance [80], although other studies did not reveal a significant influence of coat color on T*_sk_* [24,85]. Individual variations in the temperament, such as nervousness, can trigger sympathetic nerve activity resulting in vasoconstriction of skin blood vessels. This neurophysiological response may contribute to disparities in local T*_sk_*, reduced heat dissipation, and hyperthermia [93].

### 7.3. Location of T_sk_ Equipment

T*_sk_* recording at specific locations on the skin body surface shows there is a delicate equilibrium between heat influx from arterial blood, metabolic processes, local skin activities, and heat exchange with the surroundings through conduction, radiation, and evaporation. Any factors disrupting this equilibrium can induce alterations in T*_sk_*. In addition, numerous elements that influence T*_c_* during exercise concurrently impact T*_sk_*, including various performance-related factors and environmental conditions. Importantly, the precise anatomical positioning of sensors on horses for T*_sk_* measurement has been demonstrated to impact T*_sk_* outcomes [22,50,59,70,86]. For example, highest T*_sk_* readings using IRT were measured at the neck, chest, and shoulders [50,59,70]. Variations in T*_sk_* across different bodily regions (ROIs: regions of interest) such as neck, shoulder, and hip can be attributed to their varying network of blood vessels to facilitate thermal heat exchange with the surrounding environment after the competition [33,70,71]. In addition, exercise intensity (submaximal vs. maximal) and duration (prolonged vs. short) can influence T*_sk_* [22,24]. Consequently, monitoring the physiological response of the local T*_sk_* to alterations in T*_c_* during exercise over time, lacks correlation between T*_c_* and T*_sk_* [24].

### 7.4. Choice of T_sk_ Measurement Equipment

Studies with variations in surface T*_sk_* measurements may depend on the equipment type and may not be clinically relevant for horses exercising in the field [35]. For instance, a study comparing IRT and thermocouples in 12 human athletes at various time points (pre-exercise, during exercise, and post-exercise) revealed poor correlation and low reliability between the two methods [94].

## 8. Challenges and Limitations of IRT and Its Studies

It is essential to recognize that IRT sensors rely on distinct physical processes to collect data, potentially resulting in disparities in the generated data due to those variations such as type (including emissivity and the thermal window of the camera) and location of the T*_sk_* equipment (ROIs). A study of human athletes conducted in a hot environment showed close agreement between a telemetric thermistor system and a standard hard-wired thermistor system but poor agreement with a thermal camera [95]. In addition, the calibration and validation of IRT sensors before a study, particularly through comparisons with certified thermocouples in a controlled water bath, are infrequently performed.

Another limitation is differences in the methodology of IRT equipment use. For example, studies creating T*_sk_* images using a remote IRT camera positioned approximately 30 cm away from the skin surface at different body parts or of the entire body yielded diverse results [22,50,96]. The advantage of placing the IRT camera remotely away from the skin is that the local T*_sk_* will not be affected by direct contact with the equipment. However, it is important to note that the surrounding environment, such as T*_a_*, solar radiation, and coat characteristics, can influence remote T*_sk_* measurements [22,35]. In contrast, when temperature sensors are in direct contact with the skin and secured by a belt, a reliable sensor-to-skin contact is ensured which is crucial for accurate readings and minimizing the potential influence of local skin alterations and environmental factors [22,24,35]. Interestingly, one study comparing IRT T*_sk_* measurements with T*_re_* in 40 adult horses concluded that T*_sk_* was not a reliable method for determining T*_c_* [96].

In summary, IRT techniques exhibit wide variability of results in both human and equine medicine, arising from aspects such as thermographic camera placement and environmental control measures [22,23,24,35]. While consensus guidelines have been established for multiple data collection methods of human T*_sk_* using IRT [97], specific protocols for equine IRT use are still emerging [22]. Considering that additional factors such as air movement and sweating can easily and rapidly change T*_sk_* without directly affecting T*_c_*, further research is needed into reliable monitoring methods to enhance sport horse wellbeing.

## 9. Future Implications

Although monitoring T*_sk_* is generally considered straightforward, especially with the promotion of all kinds of wearable technology in the market, it is essential to accept that that there is no method of T*_sk_* monitoring that has been proven to be a reliable technique to safeguard thermo-wellbeing in horses at this point in equine research [24].

Recent advancements in human exercise research have shifted toward the application of models and algorithms that incorporate variables such as heart rate (HR) and HR variability to investigate if the T*_c_* could be reliably estimated [60,81,86,98,99]. Physiologically, HR reflects both the blood flow rate to the muscles, which is related to metabolic heat production, and the blood flow to the skin, which is related to heat loss. For example, recent studies combined continuous insulated T*_sk_* monitoring with HR monitoring in 13 and 8 human athletes exposed to hot (35 °C) and warm (25 °C) environments, respectively, and suggested a potential predictive model for T*_re_* or T*_c_* (using GI temperature pills) [81,99]. However, contrary to these findings, the HR recovery in the endurance horses in an equine study did not exhibit a direct relationship with T*_sk_* [24]. Further investigation is required to explore the potential association between T*_sk_* and HR before promoting HR and HR variability for the accurate predictive modeling of T*_c_* in equine athletes. With the ongoing development of wearable technology, future advances in temperature monitoring equipment especially targeting T*_c_* and in temperature modeling, can be expected to emerge to enable timely intervention in order to improve the welfare of the horse. Another critical motivator to advance this research field is aimed at improving the performance and health of all exercising horses.

## 10. Conclusions

The recent increased awareness of heat-related illness associated with exercise highlights the need to monitor equine core body temperature response accurately and continuously. Utilization of IRT in the realm of exercise is a relatively recent practice, and numerous fundamental different benefits and challenges warrant ongoing discussion, particularly in relation to various methodological aspects. Importantly, based on current evidence as presented, monitoring T*_sk_* during field exercise is not a valid proxy for T*_c_* monitoring. Further research could include the application of machine learning algorithms integrating T*_sk_* data in a more “holistic physiological model”. However, for that purpose, more standardized studies of equine thermoregulation specifically in the field are required.

## Figures and Tables

**Figure 1 animals-14-00136-f001:**
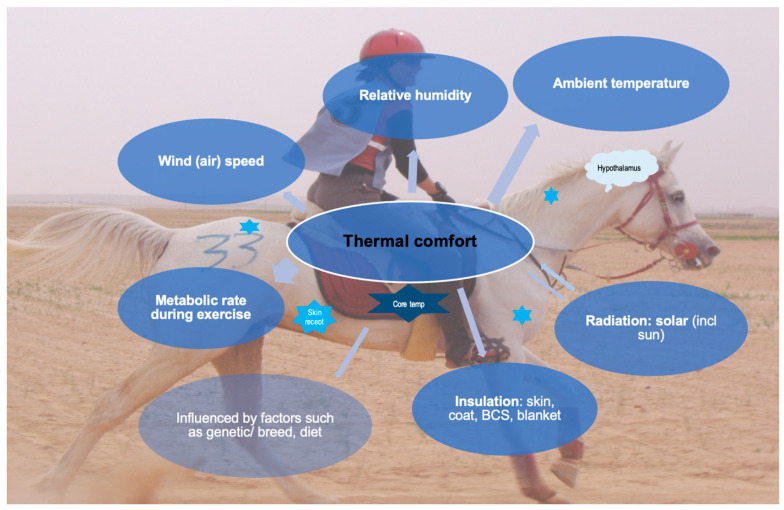
Factors affecting thermoregulation in horses and its feedback control mechanisms including thermoreceptors in the skin (spread over body, examples displayed as bright-blue 6-point stars), and the core temperature receptors (dark-blue 6-point star) which provide information to the hypothalamus (displayed as light-blue call-out), which adjusts the responses either to produce or to lose heat to ensure thermal comfort; BCS: body condition score.

**Table 1 animals-14-00136-t001:** Non-exhaustive overview of equine thermoregulatory monitoring studies measuring T*_sk_* (°C) evolvement associated with exercise (E): non-continuous data points (N/C, at point-in-time) or continuous monitoring (**C**). Aim: aim of study; IRT: infrared thermography; N: number of horses; Type: Thermo-sensor type; ROIs: regions of interest (number of locations of IRT measurement); SH: shoulder; number ICS: the location at an intercostal space; Delta T*_sk_*: T*_sk_* change pre- to post-exercise. Only essential data are presented. T*_sk_*: skin temperature; T*_CV_*: central venous blood temperature; T*_re_*: rectal temperature; T*_RA_*: right atrial blood temperature; T*_PA_*: pulmonary artery blood temperature; SET: standardized exercise testing treadmill; T*_a_*: ambient temperature; max.: maximum; min: minutes; s: seconds; field: field studies; BSA: body surface area; BM: body mass; TX: therapy; RR: respiratory rate; EHI: exertional heat illness; HH, HD, CD: hot-humid, hot-dry, cool-dry T*_a_* (table adjusted from Verdegaal et al. [24]).

Reference	Aim	N	Type	C or N/C	Compared T*_c_* Y/N	ROIs	Delta T*_sk_*	Conclusion
Verdegaal et al., 2022 [24]	E-field, C T*_sk_* compared to C T*_c_*	12	IRT device	C	Y, T*_c_* GI	1: girth at ventral thorax	11–13 °C	T*_sk_* is not proxy for T*_c_* in field
Janczarek et al., 2022 [59]	E-field, post- cooling effect	19	IRT camera	N/C	Y, T*_re_*	6: 4 legs, neck, hip	2.1–3.7 °C	T*_sk_* and delta T*_sk_* varied with body location
Brownlow and Smith, 2020 [48]	E-field, post exercise T*_sk_*	260	IRT camera	N/C: post-exercise	N	3: neck, shoulder, thorax	-	T*_sk_* > 39 °C direct post-race = higher risk for EHI development
Klous et al., 2020 [57]	E-field, pre-cooling effect	10	i-Button^®^ & glue	C	Y: C T*_re_*	2: Shoulder, rump	−3 °C	Pre-cooling (rinsing cold water for 8 min) -> T*_re_* median 0.3 °C & T*_sk_* mean −3 °C
Takahashi et al., 2020 [52]	E-SET—to 42 °C T*_PA_* post-cooling methods until T*_PA_* < 39 °C	5	IRT camera	N/C	Y: T*_PA_*	1: left thorax	N/A	When T*_PA_* 42 °C, best T*_sk_* at 17^e^ ICS > 40 °C Shower for 30 min with tap water most affective to lower T*_c_*
Wilk et al., 2020 [53]	E–field, ridden–compare BM riders	12	IRT camera	N/C	Y, T*_re_* (N/C)	7	~6 °C	Rider > 20% of horse BW -> higher T*_sk_*
Witkowska-Pilaszewicz et al., 2020 [67]	E–field, correlate with blood lactate	30	IRT camera	N/C	N	11	~max. 4 °C	>T*_sk_* at 30 min post-exercise correlated with > blood lactate (10.4 mmol/L)
Redaelli et al., 2019 [45]	E–field, intensity with T*_sk_* & cortisol	8	IRT camera	N/C	N	7	~max. 17 °C	T*_sk_* at crown of head correlated with E intensity & increased serum cortisol
Soroko et al., 2019 [46]	E–SET pre & post T*_sk_*: ridden & blood values	9	IRT camera	N/C	N	7 muscle regions	~1–2 °C	Higher T*_sk_* in ridden horses compared to non-ridden
Soroko, 2018 [44]	E–SET dynamic IRT camara	5	Dynamic use IRT camara q15s	C	N	4: SH, neck, croup, chest	~max. 4 °C	Neck was hottest ROI (34.7 °C)
Rizzo et al., 2018 [66]	E–field, Acupuncture & transport	5	IRT camera	N/C	N	6	~4 °C	Both flank T*_sk_* and T*_re_* higher post acupuncture, transport, exercise
Yarnell et al., 2014 [58]	E–SET Dry/water treadmill	7	IRT camera	N/C	N	1	~6 °C	T*_sk_* during dry treadmill assess changes in blood flow muscles
Wallsten et al., 2012 [30]	E–field, clipping & blanket in cold T*_a_*	3	Thermistor probes with sensors on skin	N/C	N	2: neck, biceps, tail	~7 °C	Unclipped & covered with blanket → higher RR & T*_re_* (38.2 °C)
Jodkowska et al., 2011 [70]	E–field, T_max_ whole BSA, jumping	35	IRT camera	N/C	Y, N/C T*_re_*	36 ROIs and 25 ROIs post- exercise	~5 °C	T*_sk_* rise highest at head, neck, trunk. T*_re_* WNL
Simon et al., 2006 [69]	E–SET time return base T*_sk_*	6	IRT camera	N/C	N	2: FL & HL	~5–6 °C	All return to base T*_sk_* in 45 min
Morgan et al., 2002 [68]	E–SET coat clipping on T*_CV_*, T*_sk_*, T*_sk_*	6	IRT thermometer	N/C	N	1	-	Clipping results in better heat loss
Geor et al., 2000 [62]	E–field, acclimation HH and CD, HD	6	Thermocouples with sticky tape	C	Y: T_PA_	1- Shaved at thorax	~2.5 °C	No difference in delta T*_sk_* (no T*_RA_* corr. ~3 °C difference)
Marlin et al., 1999 [38]	E–SET acclimation & sweating	5	Thermistor probe	C	Y: T*_RA_* N	1: tail skin (T*_PA_*, T*_re_*)	~7 °C	Onset of sweating occurred at lower T*_sk_* following acclimation
Marlin et al., 1998 [37]	E–SET cooling	5	Thermistor probe	C	Y: T*_PA_* N	2: tail skin, coat	~5 °C overall	Both T*_sk_* & T*_PA_* decreased 0.8 °C in response to cooling (6 × 30 s cold water) over 6.5 min
Marlin et al., 1996 [36]	E–SET Cool and hot T*_a_*	4	Thermistor probe	C	Y: T*_RA_* Y	1: tail skin (T*_RA_*, T*_re_*)	~6 °C	T*_sk_* follows T*_RA_* pattern
Geor et al., 1995 [61]	E–field, HH and CD, HD	5	Thermocouples with sticky tape	C	Y: T*_RA_* N	1- Shaved thorax		T*_sk_* different from T*_c_*

## Data Availability

The data presented in this study are available in article.

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
