# Peer review of "Thermoregulation during Field Exercise in Horses Using Skin Temperature Monitoring"

_animals, 2023, doi:10.3390/ani14010136_

Round 1
Reviewer 1 Report
Comments and Suggestions for Authors
Thank you for an interesting review on infrared thermography. It is needed since the measurements are quite un-invasive and simple to do so many researchers can collect a lot of data. Then it is really important to know the limitations of the data implications.
Line 13 Consider rephrasing this sentence. To me it now sounds like horses get overheated everywhere all the time.
Line 144 There is a extra point after (26)?
Line 143-153 the discussion on heat increment of feeding is not very well developed. There are studies indicating different heat production due to different feeds and nutrients. The statement that the hind gut produces enormous amounts of heat needs to have references and put in a context.
Line 214 I guess there is a letter missing in onl ?
Line 280-282 Do you think that a post-race Tsk of 39° C would be an early indicator of EHI in a cold climate?
Line 357 And hair-coat?
Line 393-402 and the intensity of the exercise in the different studies are quite different.
Author Response
We would like to thank the reviewer for providing valuable input. We appreciate the reviewer’s comments and have addressed all suggestions in itemized responses. All applied changes in the manuscript are visible in track changes in Word.
Reviewer 1
Thank you for an interesting review on infrared thermography. It is needed since the measurements are quite un-invasive and simple to do so many researchers can collect a lot of data. Then it is really important to know the limitations of the data implications.
Comment: Line 13 Consider rephrasing this sentence. To me it now sounds like horses get overheated everywhere all the time.
Response: Thank you for this suggestion. We have now modified this section (lines 13-15) into two sentences accordingly: “The process of thermoregulation is crucial for equine athletes. Excessive metabolic heat generation during exercise combined with inefficient heat dissipation, can lead to hyperthermia if not detected in time and not effectively managed.”
Comment: Line 144 There is an extra point after (26)?
Response: Thank you for this suggestion. We have now modified the sentences (lines 148-151) accordingly: “Additionally, horses are hindgut fermenters, which entails that important fermentation processes take place within the gastro-intestinal (GI) system [26], recognized for their propensity to produce an enormous amount of heat which in turn represents an additional challenge for the thermoregulatory system.
Comment: Line 143-153 the discussion on heat increment of feeding is not very well developed. There are studies indicating different heat production due to different feeds and nutrients. The statement that the hind gut produces enormous amounts of heat needs to have references and put in a context.
Response: Thank you for this suggestion. The reference 26 refers to the chapter of Ewart from the Veterinary physiology book (Ewart, S.L. Thermoregulation. In Cunningham's textbook of veterinary physiology. 6th Ed.; Bradley G. Klein Ed.; Elsevier: St Louise, USA, 2020; pp. 596 – 607) which discusses fermentation processes producing heat. However, the contribution of the fermentation process to heat generation is currently unknown, although the authors suggest this causes an additional challenge for the thermoregulatory system in horses during exercise.
Comment: Line 214 I guess there is a letter missing in onl ?
Response: Thank you for this suggestion, we have changed to only and deleted ‘a’ in line 229.
Comment: Line 280-282 Do you think that a post-race Tsk of 39° C would be an early indicator of EHI in a cold climate?
Response: Thank you for this question. The post-race Tsk of 39°C was recorded in a warm/ hot climate and we have now modified the sentence accordingly (lines 299-301): “Interestingly, an immediate post-race, high point-in-time IRT Tsk measurement > 39°C in the neck region of racehorses exercising in a warm-hot Ta has been advocated as an early indicator for increased risk for development of post-exercise EHI [48].”
Comment: Line 357 And hair-coat?
Response: Thank you for this suggestion. We have now added the coat colour to the sentence (line 377).
Comment: Line 393-402 and the intensity of the exercise in the different studies are quite different.
Response: Thank you for this suggestion. We have now modified the sentence accordingly (lines 402-404): “Environmental factors play a substantial role in shaping the dynamics of Tsk and Tc despite different exercise intensities and their interplay as these external factors have the capacity to swiftly modify Tsk without directly influencing Tc [33,65,85-88].”

Reviewer 2 Report
Comments and Suggestions for Authors
I have read and reviewed this manuscript with great interest and overall, from this reviewer's perspective, it is a study (commentary) that has been well-planned and executed. Overall it is a study with refreshingly simple wording that is easy to understand. Other strengths of the manuscript that I can highlight are the following: the introduction provides sufficient background and includes pertinent references, and the findings are correctly described. The literature is cited correctly and appropriately to the topics covered, in addition to the fact that the conclusions are supported by the results.
Nevertheless, some points must be addressed to achieve publication quality. I have left some comments hoping that they can help the authors.
General comments
L57: please add a reference.
L105: before this point, I suggest the authors generate a section called methodology. In this section, the authors will be able to write the inclusion and exclusion criteria considered in the search for information for the preparation of this manuscript. Likewise, they may expose the databases consulted as well as the keywords used in the search.
L121: I suggest the authors prepare and include a figure that shows the phenomenon of thermoregulation specifically in the horse. This will be very useful for the reader because they will be able to understand the advantages of this physiological process and the receptors that participate. Likewise, they will also understand the limitations that the equine species has in its thermoregulation.
L149: please add a reference.
L153: please add a reference.
L156: for this reason, it is important to elaborate on the figure that was suggested in L121.
L463: Please discuss the limitations and disadvantages of infrared thermography and which are factors or variables that must be controlled. For example, the emissivity of the camera, the ambient temperature, solar radiation, the color of the coat, the presence of hair, the distance between the camera and the individual, and the effect of the thermal window were evaluated, among other factors.
Author Response
I have read and reviewed this manuscript with great interest and overall, from this reviewer's perspective, it is a study (commentary) that has been well-planned and executed. Overall, it is a study with refreshingly simple wording that is easy to understand. Other strengths of the manuscript that I can highlight are the following: the introduction provides sufficient background and includes pertinent references, and the findings are correctly described. The literature is cited correctly and appropriately to the topics covered, in addition to the fact that the conclusions are supported by the results.
Nevertheless, some points must be addressed to achieve publication quality. I have left some comments hoping that they can help the authors.
Response:
We would like to thank the reviewer for providing valuable input. We appreciate the reviewer’s comments and have addressed all suggestions in itemized responses. All applied changes in the manuscript are visible in track changes in Word.
General comments
Comment: L57: please add a reference.
Response: Thank you for this suggestion. We have now added a reference accordingly: “These sudden increased ambient temperatures (Ta) are the dominant external risk factor for heat stress for human and equine athletes [1-3].
The authors have changed reference 1 (“Bouchama, A. Heatstroke: facing the threat. Crit Care Med 2006, 34, 1272-1273, doi:10.1097/01.CCM.0000208354.85490.45’) to a more recent reference of Bouchama et al. 2022 outlining the particular risk of heat waves and the risk of heat stress: “Bouchama, A.; Abuyassin, B.; Lehe, C., Laitano, O.; Jay, O.; O’Connor, F. G.; et al. Classic and exertional heatstroke. Nature Reviews Disease Primers 2022, 8, doi:10.1038/s41572-021-00334-6.”
Comment: L105: before this point, I suggest the authors generate a section called methodology. In this section, the authors will be able to write the inclusion and exclusion criteria considered in the search for information for the preparation of this manuscript. Likewise, they may expose the databases consulted as well as the keywords used in the search.
Response: Thank you for this suggestion. Our inclusion criteria were based on the following words: skin temperature, infrared thermography, continuous monitoring, thermoregulation, field exercise, horses, and athletes. We have considered generating an extra section however we have decided to refrain from this for two reasons: It’s not our goal to provide an exhaustive overview of literature with respect to the subject. Secondly, Animals description of a commentary is: articles are identified as commentaries rather than scientific papers if they contain material which is deemed to be open to interpretation and reasoned and concise arguments.
Comment: L121: I suggest the authors prepare and include a figure that shows the phenomenon of thermoregulation specifically in the horse. This will be very useful for the reader because they will be able to understand the advantages of this physiological process and the receptors that participate. Likewise, they will also understand the limitations that the equine species has in its thermoregulation.
Response: Thank you for this suggestion. We have revised figure 1 and its legend accordingly:
Figure 1. Factors affecting thermoregulation and its feedback control mechanisms including thermoreceptors in the skin (spread over body, examples displayed as bright-blue 6-point star), and the core temperature receptors (displayed as dark-blue 6-point star) which provide information to the hypothalamus (displayed as light-blue call-out), which adjusts the responses either to produce or to lose heat to ensure thermal comfort.
Comment: L149: please add a reference.
Response: Thank you for this suggestion. We have now added a reference [26] accordingly.
Comment: L153: please add a reference.
Response: Thank you for this suggestion. We have now added a reference [26] accordingly referring to a general explanation for endotherms requiring maintenance of body temperature using a food with a high metabolic rate to provide heat.
Comment: L156: for this reason, it is important to elaborate on the figure that was suggested in L121.
Response: Thank you for this suggestion. We have modified Figure 1 accordingly.
Comment: L463: Please discuss the limitations and disadvantages of infrared thermography and which are factors or variables that must be controlled. For example, the emissivity of the camera, the ambient temperature, solar radiation, the color of the coat, the presence of hair, the distance between the camera and the individual, and the effect of the thermal window were evaluated, among other factors.
Response: Thank you for this suggestion. The authors agree that there are limitations and disadvantages when using infrared thermography. Please refer to the sections and lines 453-471 in the manuscript section 8: “8. Challenges and limitations of IRT and its studies”. The limitations and disadvantages are address more specifically in the following sections:
- Emissivity and thermal window of the camera is added to lines 461-464 accordingly: “It is essential to recognize that IRT sensors rely on distinct physical processes to collect data, potentially resulting in disparities in the generated data due to those variations such as type (including emissivity and thermal window of the camera) and location of the Tsk”
- Lines 475-480 discuss the influence of the surroundings and is changed accordingly: “However, it is important to note that the surrounding environment such as Ta, solar radiation, coat characteristics, can influence remote Tsk measurements [22,35]. In contrast, when temperature sensors are in direct contact with the skin and secured by a belt, a reliable sensor-to-skin contact is ensured which is crucial for accurate readings and minimizing the potential influence of local skin alterations and environmental factors such as Ta and solar radiation [22,24,35].”
- In addition, section 7.1 lines 280-283 discussed location of skin temperature measurement: “For example, a study involving eight endurance horses analyzed the association between endurance training intensity and Tsk evolvement using IRT cameras focused at different locations on the body.” And in addition, lines 469-475 discuss the importance of the camera distance: “For example, studies creating Tsk images using a remote IRT camera positioned approximately 30 cm away from the skin surface at different body parts or the entire body yielded diverse results [22,50,96]. One study comparing IRT Tsk measurements with Tre in 40 adult horses concluded that Tsk was not a reliable method for determining Tc [96]. The advantage of placing the IRT camera remotely away from the skin is that the local Tsk will not be affected by direct contact with the equipment.”
- The suggestion adding coat characteristics is changed accordingly (lines 427-430): “Various skin-related factors such as sweating rate, skin thickness, blood vessel density, hair coat characteristics, presence of hair, clipping practices, coat color and anhidrosis (inability to sweat) all influence the efficacy of their sweat evaporation and the Tsk measurement [27,30,33,53,54,56,65,80,91,92].” In addition, lines 433-436 further discuss the influence of coat color: “Additionally, coat color may hold some relevance [80] however, other studies did not reveal a significant influence of coat color on Tsk [24,85]. Individual variations in the temperament, such as nervousness, can trigger sympathetic nerve activity resulting in vasoconstriction of skin blood vessels.”

Reviewer 3 Report
Comments and Suggestions for Authors
This manuscript details the limitations of using skin surface temperatures by performing a literature review – likely as part of a dissertation. The work provides convincing support for the idea that just monitoring skin surface temperature is not an adequate measure for determining whether health concerns may be present for horses.
There are several items that this reviewer believes need to be addressed or changed. In the Introduction (lines 50-54), global climate change is charged with making the incidence of heat stress and exertional heat illness increase. While climate change is often used to justify research, it does not seem warranted for this paper. According to Climate.gov, the rate of warming has increased 0.18 degrees per decade since 1981. This increase would seemingly have only a minor impact on the thermal load of a horse compared to just changes in temperature that typically occur throughout the day or with changes in season. (Compared to those changes, the amount that can be attributed to the rate of warming is negligible.) Again, I understand many people try to use climate change to justify their research or get funding for their work, it just does not seem like it is good justification for doing this work and I would like to see it removed from the Introduction. Your work stands alone without trying to use climate change to justify it – particularly when it likely would have a minor impact compared to all other factors that have a much greater impact. Even a horse’s color would have a much greater impact than the 0.18 degrees per decade increase since 1981.
In line 144, there seems to be some incomplete sentences (“the gastro-intestinal system [26]. are recognized”)
In lines 172-173, I believe this sentence needs to be reworded. I would suggest breaking the sentence into two. For instance, “influences exercise intensity. For example, the presence...”
Once again, I do not believe using climate change to justify your work is necessary (or appropriate). This was done in lines 372-374. According to the Guinness Book of World Records (https://www.guinnessworldrecords.com/world-records/greatest-temperature-range-in-day#:~:text=The%20greatest%20temperature%20variation%20in,8%20a.m.%20on%2015%20Jan.), the greatest temperature variation in a single location in a 24-hour period is 57.2 degrees C in Loma, Montana in January or 1972 when it went from -47.7 degrees C to 9.4 degrees C. In short, weather changes and so do temperatures. Trying to say it is “Due to climate change” negates the fact that it has always happened and is something that can provide a challenge to a horse. There is no need to try and justify it by claiming it is “due to climate change”, thus, please remove that part of the sentence.
Line 471-472. This sentence is confusing “that no method of Tsk monitoring is not a reliable technique”. Possibly reword to “that there is no method of Txk monitoring that has been proven to be a reliable technique”.
Lines 489-490. Please reword. That does not appear to be a complete sentence.
Author Response
This manuscript details the limitations of using skin surface temperatures by performing a literature review – likely as part of a dissertation. The work provides convincing support for the idea that just monitoring skin surface temperature is not an adequate measure for determining whether health concerns may be present for horses.
There are several items that this reviewer believes need to be addressed or changed.
Response
We would like to thank the reviewer for providing valuable input. We appreciate the reviewer’s comments and have addressed all suggestions in itemized responses. All applied changes in the manuscript are visible in track changes in Word.
Comment: In the Introduction (lines 50-54), global climate change is charged with making the incidence of heat stress and exertional heat illness increase. While climate change is often used to justify research, it does not seem warranted for this paper. According to Climate.gov, the rate of warming has increased 0.18 degrees per decade since 1981. This increase would seemingly have only a minor impact on the thermal load of a horse compared to just changes in temperature that typically occur throughout the day or with changes in season. (Compared to those changes, the amount that can be attributed to the rate of warming is negligible.) Again, I understand many people try to use climate change to justify their research or get funding for their work, it just does not seem like it is good justification for doing this work and I would like to see it removed from the Introduction. Your work stands alone without trying to use climate change to justify it – particularly when it likely would have a minor impact compared to all other factors that have a much greater impact. Even a horse’s color would have a much greater impact than the 0.18 degrees per decade increase since 1981.
Response: Thank you for this suggestion. We have now modified the sentence accordingly (lines 54-59): “Climate change is a generally accepted phenomenon, and the scientific community is warning the world about its long-term consequences, both for human and animal health [1-3]. However, more acute changes in environmental conditions, particularly heat waves, entail that athletes may need to perform under circumstances to which they are not acclimatized. These sudden increased ambient temperatures (Ta) are the dominant external risk factor for heat stress for human and equine athletes [1-3].
The authors have changed reference 1 (“Bouchama, A. Heatstroke: facing the threat. Crit Care Med 2006, 34, 1272-1273, doi:10.1097/01.CCM.0000208354.85490.45’) to a more recent reference of Bouchama et al. 2022 outlining the particular risk of heat waves and the risk of heat stress: “Bouchama, A.; Abuyassin, B.; Lehe, C., Laitano, O.; Jay, O.; O’Connor, F. G.; et al. Classic and exertional heatstroke. Nature Reviews Disease Primers 2022, 8, doi:10.1038/s41572-021-00334-6.”
Comment: In line 144, there seems to be some incomplete sentences (“the gastro-intestinal system [26]. are recognized”)
Response: Thank you for this suggestion. We have now modified the sentences (lines 147-153) accordingly: “Additionally, horses are hindgut fermenters, which entails that important fermentation processes take place within the gastro-intestinal (GI) system [26], recognized for their propensity to produce an enormous amount of heat which in turn represents an additional challenge for the thermoregulatory system in horses. It is quite plausible that the type of dietary load (composition of intestinal content) inside the GI system of a horse performing exercise has its impact on core body temperature evolution at that time point [26].
Comment: In lines 172-173, I believe this sentence needs to be reworded. I would suggest breaking the sentence into two. For instance, “influences exercise intensity. For example, the presence...”
Response: Thank you for this question and suggestion. We have now modified the sentence (lines 183-185) accordingly: “Additionally, the terrain itself influences exercise intensity. For example, the presence of slopes and descents in a competition course as well as varying properties of the surface over the course, such as sand or mud, are challenging for optimal exercise capacity [43].
Comment: Once again, I do not believe using climate change to justify your work is necessary (or appropriate). This was done in lines 372-374. According to the Guinness Book of World Records (https://www.guinnessworldrecords.com/world-records/greatest-temperature-range-in-day#:~:text=The%20greatest%20temperature%20variation%20in,8%20a.m.%20on%2015%20Jan.), the greatest temperature variation in a single location in a 24-hour period is 57.2 degrees C in Loma, Montana in January or 1972 when it went from -47.7 degrees C to 9.4 degrees C. In short, weather changes and so do temperatures. Trying to say it is “Due to climate change” negates the fact that it has always happened and is something that can provide a challenge to a horse. There is no need to try and justify it by claiming it is “due to climate change”, thus, please remove that part of the sentence.
Response: Thank you for this suggestion. The authors have outlined their reasoning in an earlier response with the main argument in particular the heat waves association with shifts in weather patterns due to long-term climate change which entail that athletes may need to perform under circumstances to which they are not acclimatized. These sudden increased ambient temperatures (Ta) are the dominant external risk factor for heat stress for human and equine athletes [1-3]. In particular reference from Bouchama et al. outlines the risk of heat stress due to sudden heat waves.
Comment: Line 471-472. This sentence is confusing “that no method of Tsk monitoring is not a reliable technique”. Possibly reword to “that there is no method of Txk monitoring that has been proven to be a reliable technique”.
Response: Thank you for this excellent suggestion. We have now modified the sentence (lines 492-495) accordingly: “Although monitoring Tsk is generally considered straightforward, especially with the promotion of all kinds of wearable technology in the market, it is essential to accept that that there is no method of Tsk monitoring that has been proven to be a reliable technique to safeguard thermo-wellbeing in horses at this point in equine research [24].”
Comment: Lines 489-490. Please reword. That does not appear to be a complete sentence.
Response: Thank you for this suggestion. We have now modified the sentences (lines 508-512) accordingly: “With the ongoing development of wearable technology, future advances in temperature monitoring equipment especially targeting Tc and temperature modelling can be expected to emerge to enable timely intervention in order to improve the welfare of the horse. Another critical motivator to advance this research field is aimed at improving the performance and health of all exercising horses.”
